# GALA: a computational framework for de novo chromosome-by-chromosome assembly with long reads

**Mohamed Awad**[1] **& Xiangchao Gan** [1,2] ✉

High-quality genome assembly has wide applications in genetics and medical studies. However, it is still very challenging to achieve gap-free chromosome-scale assemblies using current workflows for long-read platforms. Here we report on GALA (**Ga**p-free **l**ong-read **A**ssembly tool), a computational framework for chromosome-based sequencing data separation and de novo assembly implemented through a multi-layer graph that identifies discordances within preliminary assemblies and partitions the data into chromosome-scale scaffolding groups. The subsequent independent assembly of each scaffolding group generates a gap-free assembly likely free from the mis-assembly errors which usually hamper existing workflows. This flexible framework also allows us to integrate data from various technologies, such as Hi-C, genetic maps, and even motif analyses to generate gap-free chromosome-scale assemblies. As a proof of principle we de novo assemble the *C. elegans* genome using combined PacBio and Nanopore sequencing data and a rice cultivar genome using Nanopore sequencing data from publicly available datasets. We also demonstrate the proposed method's applicability with a gap-free assembly of the human genome using PacBio high-fidelity (HiFi) long reads. Thus, our method enables straightforward assembly of genomes with multiple data sources and overcomes barriers that at present restrict the application of de novo genome assembly technology.

De novo genome assembly has wide applications in plant, animal, and human genetics. However, it is still very challenging for long-read platforms, such as Nanopore and PacBio (Pacific Bioscience), to provide chromosome-scale assemblies[1,2]. To date, numerous de novo assembly tools have been developed to obtain longer and more accurate representative sequences from raw sequencing data[3–5]. In most studies, however, assemblies by these tools comprise hundreds or even thousands of contigs. To produce chromosome-scale assembly, various information sources, such as Hi-C, genetic maps, or a reference genome, have been increasingly used to anchor contigs into big scaffolds[6,7]. As a consequence, the final genome assembly usually contains numerous gaps, and sometimes, is also plagued with mis-assemblies, as reported in ref. [8].

Gaps and mis-assemblies in a genome assembly can seriously undermine genomic studies. For example, a lot of sequence alignment tools have much lower performances when query sequences contain gaps[9,10]. In intraspecific genome comparisons, large gaps not only significantly increase the possibility of failure to detect long structure variants, but also produce inaccurate results of gene annotation[11,12]. Moreover, gaps and mis-assemblies have been reported to account for a large number of gene model errors in existing genome assembly studies[13,14].

[1]Max Planck Institute for Plant Breeding Research, Department of Comparative Development and Genetics, Carl-von-Linné-Weg 10, 50829 Köln, Germany. [2]State Key Laboratory for Crop Genetics and Germplasm Enhancement, Bioinformatics Center, Academy for Advanced Interdisciplinary Studies, Nanjing Agricultural University, 210095 Nanjing, China. ✉e-mail: gan@mpipz.mpg.de

Plant and animal genomes usually contain multiple chromosomes. Most past and current genome studies sequence all chromosomes together. There are two main drawbacks for this pooled sequencing. The first is that extra computational resources are required for data analyses. For example, computational loads and storage may increase exponentially with data size for some de novo assembly algorithms, and alignment algorithms can be ten times faster if reads are only aligned to a specific chromosome rather than to the whole genome. The second drawback is that the sequencing data, especially those from repetitive regions or mobile elements, may interfere with each other. To allow single-chromosome sequencing, the chromosome flow-sorting technique has been proposed. It has been successfully applied to human, wheat[15] and some other genome studies[16,17]. However, chromosome sorting techniques required highly complicated protocols to resolve the optical properties, e.g., light scatters and fluorescence of the target chromosomes[18], making it highly expensive, time-consuming and labour-intensive, thus limiting its applications.

In this study, we report on GALA (**Ga**p-free **l**ong-read **A**ssembly tool), a computational framework for chromosome-based sequencing data separation and assembly. GALA is implemented through a multi-layer computer graph (Fig. 1) and separates two steps: firstly, it clusters raw reads and contigs from preliminary assemblies into multiple scaffolding groups, each representing a single chromosome (sometimes a chromosome arm); secondly, it assembles each scaffolding group from raw reads. Moreover, our method can also exploit the information derived from Hi-C data to obtain chromosome-scale scaffolding groups in studies even with a complicated genome structure or those with low sequencing quality. Of note is that our method can be easily extended to incorporate other sources of information such as genetic maps or even a reference genome. Here, we show the utility of GALA by gap-free and chromosome-scale assemblies of PacBio or Nanopore sequencing data from two publicly available datasets for which the original assembly contains large gaps and a number of unanchored scaffolds. Notably, our method significantly outperforms existing algorithms in both datasets. Finally, we also demonstrate the application of our method to assemble a human genome with the help of a reference genome using PacBio high-fidelity (HiFi) long reads.

## Results

### Overview of the GALA framework

GALA exploits information from multiple de novo assembly tools and raw reads, as well as other information sources, such as Hi-C, genetic maps, or even a reference genome, if they exist. In GALA, various de novo assembly tools are selected first to create preliminary assemblies. These preliminary assemblies and raw reads are then aligned against each other. We use a multi-layer computer graph to model the GALA, with each assembly encoded as one layer, together with an extra layer representing the raw reads. Inside each layer, a contig (or a read in the raw-read layer) is encoded as a graph node. GALA browses through the reciprocal alignments and other information sources and creates two types of edges. Any information between preliminary assemblies or raw reads is recorded as an inter-layer edge. Inside each layer, for two nodes, if extra information sources, e.g., a Hi-C sequencing data alignment or a linkage map, supports that they both belong to the same chromosome, an intra-layer edge is created between them (Fig. 2 and Supplementary Fig. 1).

Depending on the sequencing quality and complexity of the genome structure, existing assembly tools usually exhibit different performances in terms of the number of misassembled contigs and N50. To prevent the spread of errors, we developed a mis-assembly detection module (MDM). This module works on layers of preliminary assemblies by estimating the probability of mis-assemblies, i.e., chimeric contigs, based on the contradictory inter-layer edges, and splitting those nodes containing highly likely mis-assemblies to resolve discordances in the computer graph ("Methods"). After removing contradictory inter-layer links, the contig-clustering module (CCM) pools all nodes within the multi-layer graph which can reach each other through inter- and intra-layer edges into the same scaffolding group, in many cases each representing a chromosome ("Methods"). In several experiments, we identified orphan scaffolding groups which consist of a single contig from one layer. Interestingly, most of them come from external sources such as bacterial or sample contamination.

The successful partitioning of existing preliminary assemblies and raw reads into separate scaffolding groups allows us to essentially perform the de novo assembly on each scaffolding group or chromosome separately, rather than on the complete genome with multiple chromosomes. We called this strategy the de novo chromosome-by-chromosome assembly. With this strategy, chromosomes can be assembled in parallel then pooled together for the complete genome. Detailed comparisons indicated that chromosome-by-chromosome assembly provides better performance than straightforward assembly of a mixture of reads from multiple chromosomes, i.e., the original sequencing data, especially for the repetitive fragments in terms of contiguity. We further investigated whether this improvement has

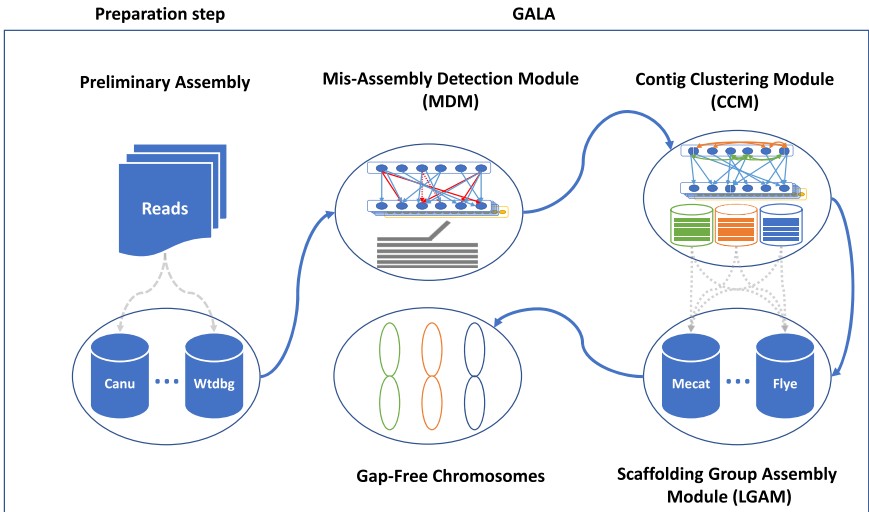

**Fig. 1 | Overview of GALA.** After de novo assembling with various tools, preliminary assemblies and raw reads are encoded into a multi-layer computer graph. Mis-assemblies are identified with MDM by browsing through the inter-layer information. The split nodes are clustered into multiple linage groups by the CCM. Each scaffolding group is assembled independently using SGAM to achieve the final gap-free sequences of chromosomes.

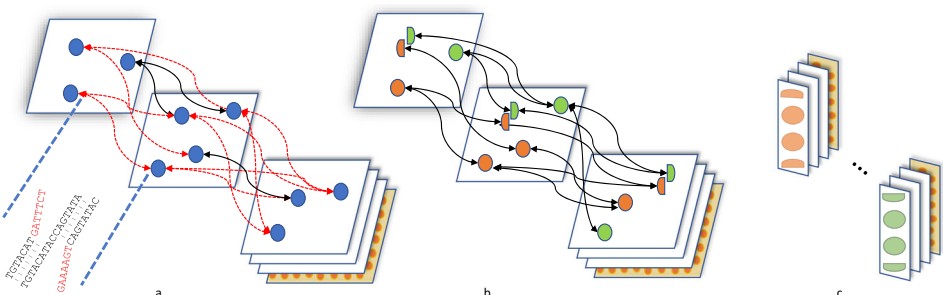

**Fig. 2 | Illustration of a multi-layer computer graph in GALA. a** The preliminary assemblies and raw reads are aligned against each other and encoded into a multi-layer graph. Conflicted alignments are encoded with edges in red. **b** The conflicted alignments are removed iteratively by splitting the nodes involved and new edges are assigned accordingly. The procedure stops only after all conflicted alignments in the system have been resolved. **c** Nodes connected by edges are clustered into scaffolding groups.

something to do with the read correction procedure. To our surprise, the improvement of read correction with chromosome-by-chromosome analysis is negligible. We also tested a scaffolding-based strategy, where the consensus assembly for each chromosome is obtained by merging the assembled contigs within the scaffolding group without working on raw reads. However, in many cases, the fast mode generated gapped assemblies, thereby highlighting the distinct advantage of the chromosome-by-chromosome assembly strategy over existing tools.

### *Caenorhabditis elegans* genome assembly

We used a publicly available dataset for *Caenorhabditis elegans* VC2010. The dataset was generated on the PacBio platform with a 290× coverage along with an extra 140× coverage of Nanopore sequences[19,20]. As most existing assembly tools do not support pooled sequencing data from PacBio and Nanopore platforms, we used both datasets separately to generate preliminary assemblies (Supplementary Fig. 2). Preliminary assemblies were generated using Canu, Flye, Mecat2/Necat, Miniasm, and Wtdbg2 ("Methods"). Among all our preliminary assemblies, the one produced by PacBio-Flye showed the smallest number of contigs, with 41 contigs for 102 Mbp of overall sequences.

We applied GALA to the raw reads and the preliminary assemblies. The number of discordances in each preliminary assembly derived by the MDM algorithm ranged from 0 to 19. After resolving the discordances through the node-splitting operation, GALA modelled the input into 14 independent scaffolding groups. Seven of them contain a very small amount of sequencing data and apparently come from short continuous contigs. Among them, four contigs are from bacterial contamination or organelle DNA and two of them can be pooled into seven large scaffolding groups using Nanopore sequencing data. The remaining one contains a telomeric repetitive motif. We then performed telomeric motif analyses for the seven large scaffolding groups. Four of them contain complete chromosomes. Two groups contain the telomeric repetitive motif at one end and apparently come from two arms of the same chromosome and one group misses the telomeric repetitive motif at one end. We thus were able to merge the nine autosome scaffolding groups further into six ones (Supplementary Fig. 3 and "Methods"). Of note is that the subsequent de novo assembly of each scaffolding group generated gap-free complete sequences for all six chromosomes. The N50 of the GALA assembly shows 4.2-fold and 9.6-fold increases comparing to the highest and lowest N50 of the preliminary assemblies, respectively (Supplementary Tables 1 and 2). Moreover, fewer collapsed regions with significant shorter length have been observed in GALA assembly when comparing to the preliminary assemblies (Supplementary Table 3).

We polished our assembly using PacBio and Illumina short reads and then compared it to the published VC2010 assembly[19] and the N2 reference genome[21]. Note that the VC2010 sample is derived from the N2

reference sample and is widely used as a substitute to the reference strain which is unavailable nowadays[22,23], and their assemblies are supposed to be very close. The evaluation from Busco 3.0.0 indicated that our assembly successfully assembled two more genes, which are absent in both reference and the published assembly. Furthermore, the alignment of Illumina short reads against our assembly also reveals a better alignment rate as well as fewer variants (Table 1 and Supplementary Fig. 4).

We performed additional analyses to test the performance of our assembly using the Hi-C dataset generated by the same research group[19]. No discordances were revealed by aligning the Hi-C data against our assembly using BWA-MEM, then detecting the discordances using Salsa[24]. Salsa also supported the merging of two scaffolding groups suggested by the telomeric motif analyses in our assembly. For comparison, we also applied Salsa with Hi-C data to the best preliminary assembly from Flye with PacBio data. This Flye/Hi-C assembly contains seven scaffolds and 14 unanchored contigs after excluding those from sample contamination. We observed 17 spanned gaps in the Flye/Hi-C assembly, with the two largest gaps being 495 Kbp and 159 Kbp (Fig. 3). Furthermore, we aligned the raw PacBio reads to different assemblies and examined the distribution of the depth-of-coverage across the genome (Supplementary Fig. 5). Apart from containing no gaps, the GALA assembly shows comparable performance to the VC2010 assembly in terms of assembly error in repetitive regions.

**Table 1 | The assembly performance evaluation of GALA with Busco scores and statistics of alignment of Illumina short reads compared to the N2 reference genome[21] and the VC2010 assembly[19]**

| | N2 reference genome | VC2010 assembly | GALA assembly |
|---|---|---|---|
| Assembly length | 100,286,401 | 102,092,263 | 102,301,025 |
| Number of contigs | 7 | 7 | 7 |
| Busco complete | 968/982 | 968/982 | 970/982 |
| Busco duplicated | 6/982 | 6/982 | 6/982 |
| Busco fragmented | 8/982 | 8/982 | 6/982 |
| Busco Missing | 6/982 | 6/982 | 6/982 |
| QV | 36.4155 | 36.0716 | 36.2818 |
| Mapped reads | 130,604,410 | 130,639,345 | 130,652,108 |
| Unmapped reads | 4,568,540 | 4,533,605 | 4,520,842 |
| Variants | 17,385 | 14,839 | 14,169 |
| SNPs | 16,179 | 14,167 | 13,701 |
| Deletions | 412 | 282 | 124 |
| Insertions | 794 | 390 | 344 |
| Indels | 1206 | 672 | 468 |

The Busco scores are computed using Busco V.3.0.0 with nematoda odb9 database. The QV scores are calculated using merqury reference free assessment tool.

### *Oryza sativa* genome assembly

We assembled *Oryza sativa* circum-basmati landrace Dom Sufid (sadri) using a publicly available dataset with GALA. The dataset contains 42.7 GB Nanopore sequencing data, equivalent to 56x coverage of the rice genome with 12 chromosomes[25]. Firstly, we used the Canu self-correction and trimming module to correct the raw reads, and produced a preliminary assembly with corrected reads. Flye, Miniasm and Wtdbg2 were used to generate six preliminary assemblies using raw and corrected reads respectively. In addition, Necat produced a preliminary assembly from the raw reads.

GALA analyses on the preliminary assemblies highlighted a number of discordances in each preliminary assembly, which ranged from 0 to 2. The input was rectified and separated into 16 independent scaffolding groups. Among them, one was from the mitochondrial genome and one from the chloroplast genome. The remaining 14 scaffolding groups represent ten chromosomes and four chromosome arms. For these four chromosome-arm scale scaffolding groups which represent two chromosomes (Chr2 and Chr11), there are only three possible combinations. We run scaffolding group assembly module (SGAM) on each of combination, only one produced continuous telomere-to-telomere pseudomolecules for both chromosomes. The scaffolding group assembly on the ten chromosome-scale scaffolding groups generated ten gap-free complete contigs. In total, our final assembly produced 12 gap-free complete chromosome sequences. GALA assembly has N50 of 30.7 Mbp and L50 of 6, in contrast to N50 of

24.1 Mbp and L50 of 7 from the best complete preliminary assembly, respectively (Supplementary Tables 1 and 2). Moreover, GALA assembly also contains the smallest size of collapsed regions compared to the preliminary assemblies (Supplementary Table 3).

In addition, our assembly showed an inversion on Chr6 compared to the reference genome of *Oryza sativa* L. ssp. *japonica* cv. Nipponbare[26,27]. This inversion was reported in the circum-basmati genome study but the previous Dom Sufid assembly cannot produce a gap-free complete sequence for this region[25]. The de novo assembly of *Oryza sativa* using GALA significantly improved the previous Dom Sufid assembly which was generated through a reference-guided scaffolding method[25] (Supplementary Fig. 6).

### Human genome assembly

We next assembled a human genome using high-fidelity (HiFi) long reads generated by PacBio using the circular consensus sequencing (CCS) mode[28]. For simplicity, we used the published preliminary de novo assembly by HiCanu[28] (3.28 GB overall) and the current human reference genome GRCh38.p13 as input for GALA. The raw reads and the input HiCanu preliminary assembly are partitioned by the contig-clustering module (CCM) of GALA. Here, the CCM only serves as a raw-read separation tool to enable subsequent chromosome-by-chromosome de novo assembly. Both information from the input reference genome, which could be from a close relative thus different from the query genome, and information from the preliminary assembly of the query genome, were used for raw-read separation. GALA revealed 23 independent scaffolding groups and assembled them one-by-one. Notably, when assembling scaffolding groups, we used two software tools, namely HiCanu and Hifiasm, and they provided significantly different assemblies in terms of the length of sequences. Taking Chromosome 17 as an example, HiCanu assembled its scaffolding group into three contigs with a total length of 83.2 Mb (40 Mb, 24.7 Mb, and 18.5 Mb). In contrast, Hifiasm produced one single telomere-to-telomere contig of a total length of 82.1 Mb. To resolve this, we aligned the raw HiFi reads to both assemblies and examined the distribution of the depth-of-coverage. We selected the better genome assembly by taking into account the number of assembly errors as well as gaps. The comparison between our GALA assembly and the published assembly can be found in Fig. 4a and Supplementary Fig. 7. Overall, our assembly comprised of 38 continuous contigs, including seven telomere-to-telomere gap-free pseudomolecular sequences (3, 7, 10, 11, 16, 17 and 20), four near-complete chromosomes (5, 8, 12 and 19) each with a small telomeric fragment unanchored, and four chromosomes (4, 6, 9 and 18) with gapped centromeric regions. Note that we only assembled the long arms of the five acrocentric chromosomes (13, 14, 15, 21 and 22) since the sequencing and assembly of their *p* arms are too challenging as they

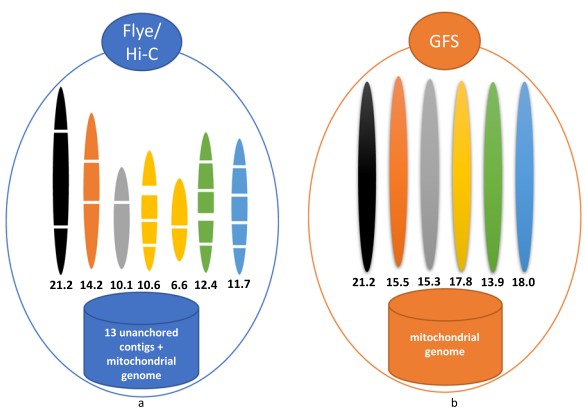

**Fig. 3 | Comparison of Flye assembly with Hi-C scaffolding and GALA assembly of long reads of the *C. elegans* genome. a** The Flye assembly with Hi-C scaffolding contains numerous gaps and 13 unanchored contigs in the assembly. **b** GALA produces gap-free assembly for each chromosome. Note this is not a fair comparison since GALA did not use Hi-C data in this assembly.

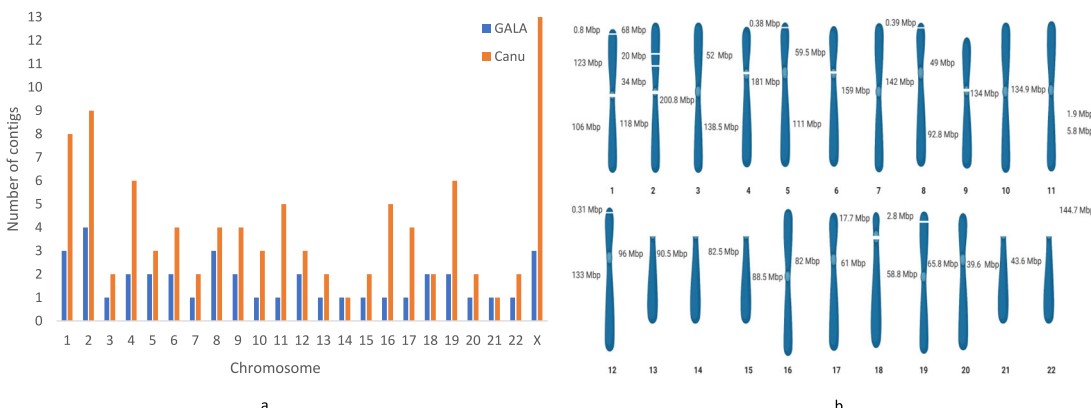

**Fig. 4 | Human genome assembly by GALA. a** Comparison of the number of contigs in assemblies by Canu and GALA. **b** A cartoon presentation of each chromosome assembled by GALA with the lengths of contigs labelled, created with BioRender.com.

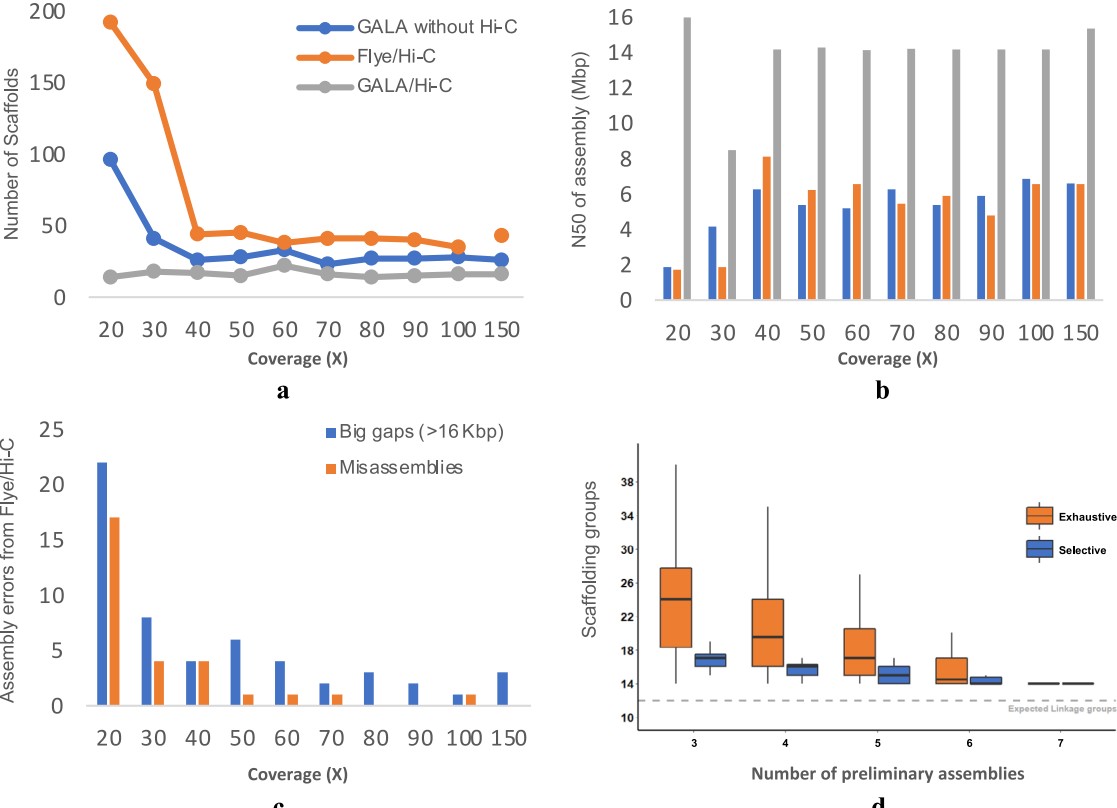

**Fig. 5 | The assembly performances of GALA changes with the coverages of sequencing data and with the number and the quality of preliminary assemblies.** To investigate the effect of sequencing coverage, three assembly procedures have been tested using *C. elegans* PacBio sequencing data: GALA without Hi-C data, Flye/Hi-C and GALA/Hi-C. The assembly performances are evaluated in terms of **a** the number of scaffolds, **b** N50, and **c** the number of big gaps (>16 Kbp) and mis-assemblies. In (**c**), only the number of gaps and mis-assemblies for Flye/Hi-C have been shown for simplicity, as GALA shows overwhelming advantage and only one mis-assembly has been identified in the assembly by GALA without Hi-C data with 30× coverage of sequencing data. In (**d**), the number of scaffolding groups obtained by GALA changes significantly with the number and the quality of preliminary assemblies used in the Oryza sativa circum-basmati landrace Dom Sufid genome assembly with the Nanopore sequencing data. Here *n* = 35, 35, 21, 7 and 1 for the exhaustive strategy, and *n* = 15, 20, 15, 6, and 1 for the selective strategy, respectively. The median (line), 1st and 3rd quartiles (bounds of the box), minimum and maximum (whiskers) are shown in the box plot.

are almost all missing in both the reference genome and the published assembly. The successful assembly of the human genome indicates that GALA works efficiently on PacBio HiFi data.

Our human genome assembly is depicted chromosome-by-chromosome in Fig. 4b. Here, two chromosomes are of key interest: Chromosome 11 and Chromosome X. In the reference genome GRCh38.p13 and also the published HiCanu assembly, Chromosome 11 has several gaps and unanchored contigs. It is considered as one of the chromosomes with the highest density of genes linked with genetic diseases[29]. GALA successfully assembled this chromosome into a single contig free of gaps of a total length of 134.9 Mbp with two telomeric regions at both ends. In contrast, one telomeric region is missing in GRCh38.p13. The length of the assembled contig by GALA and the alignment depth profile shows that the quality of the assembly of the chromosome is highly comparable with the T2T assembly (Supplementary Fig. 7). The second example is chromosome X, whose assembly is regarded as highly challenging and a lot of extra data and laborious efforts have been devoted to this in a recent paper[30]. Our assembly only contains two short gaps (about 0.75 Kbp and 1.8 Kbp) compared to the published one. The golden-standard T2T assembly is 1.7 Mbp longer compared to GALA assembly. Further investigation indicates that GALA assembly has collapsed the higher-order repeat (HOR) region with the chromosome-specific alpha satellite array DXZ1 (GALA assembled it to ~1.1 Mbp out of ~2.8 Mbp). The region was assembled in the T2T project with a lot of extra data and manual curation. Comparison between the raw reads aligned to DXZ1 of the T2T v1.0 assembly and the reads in GALA's Chromosome X scaffolding

group, shows only a single read in the former is missing from the latter, indicating that GALA's reads separation module works efficiently. But existing de novo assembly software tools we used for de novo assembly of the scaffolding group unfortunately cannot resolve the DXZ1 regions. In Supplementary Figs. 8a–c and 9, we demonstrate the performance of gap closing by GALA with three examples and the statistics of gaps in the euchromatin regions in the reference genome GRCH38.p13 which have been successfully assembled in the GALA assembly when evaluated using the T2T assembly as the benchmark. Apart from gaps in the GALA assembly mentioned previously, the remaining inconsistent regions between the GALA assembly and the T2T assembly are copy number differences of repetitive sequences and most of them are located in centromeric regions and telomeric regions. In Supplementary Fig. 8e–g, we show three examples in carton. In Supplementary Fig. 10, we plot the length of the centromeric regions in the GALA assembly and the T2T assembly.

In the above assembly of CHM13 by GALA, the reference genome was used to help separate raw read into scaffolding groups. One might wonder whether this would lead to a vulnerability that plagues traditional *reference-guided* assemblies or scaffolding. It has been reported that traditional *reference-guided* assemblies suffer from short-length assembly errors and mis-scaffolding because of reference biases and chromosomal rearrangements among different strains and cell lines, as well as errors of sequence alignment[31–33]. In addition, *reference-guided* assembly leads to missing sequences in highly divergent regions[32]. Fortunately, GALA can avoid both problems. Firstly, GALA only uses the reference genome to cluster contigs from the

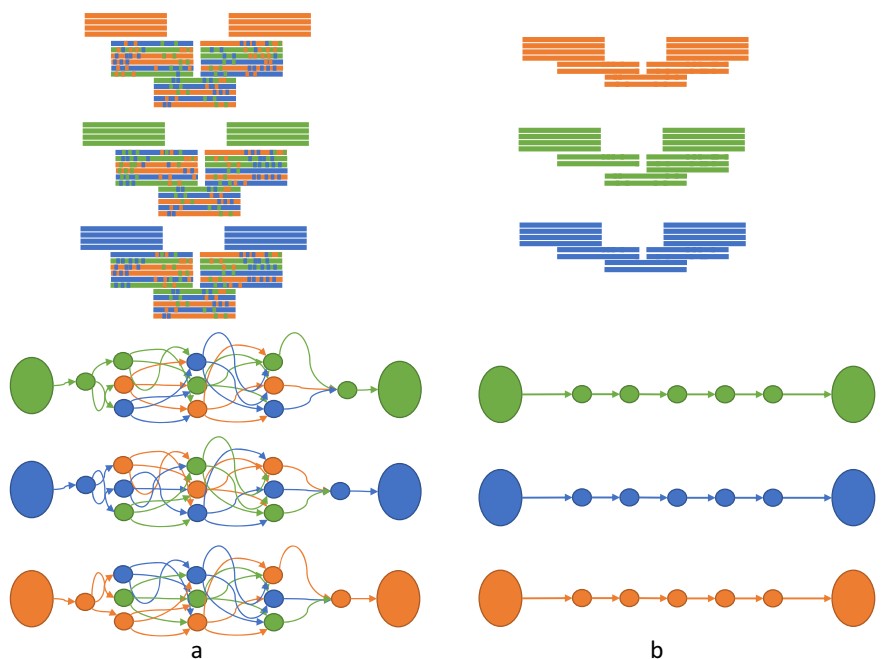

**Fig. 6 | Comparison of the overlap graphs used by Miniasm during assembly of a region in the *C. elegans* genomes when the chromosome-by-chromosome strategy is applied or not. a** In the whole genome assembly mode, the overlap graph used by Miniasm contains numerous edges and extra effort is needed to collapse edges. **b** The chromosome-by-chromosome assembly allows a linear overlap graph to be derived by Miniasm in the same region.

preliminary assembly and raw reads, so in this respect the reference functions more like a genetic map, and is largely insensitive to sequence variation between the query genome and the reference. Moreover, the subsequent de novo assembly of scaffolding groups prevents assembly errors and mis-scaffolds. For example, if raw reads are mistakenly placed into the same scaffolding group, this leads to assembly fragmentation rather than other types of errors. Secondly, GALA's scaffolding groups contain contigs from the preliminary assembly, so unique and highly divergent regions absent from the reference would not be missed out when aligning raw reads to scaffolding groups. For comparison, we performed the *reference-guided* scaffolding of the HiCanu preliminary assembly using Ragoo[34] and gap-filled it using PBJelly[35]. Ragoo scaffolded ~12 Mbp of centromeric and pre-centromeric sequences of Chr9 to Chr4 (Supplementary Fig. 11) with big gaps. In contrast, GALA clustered and assembled the reads from highly similar centromeric regions and constructed two continuous contigs in the two regions.

### Effect of the sequencing depth and the number of preliminary assemblies on the performance of GALA

We next investigated how the performance of GALA changes depending on the sequencing depth. We subsampled the original *C. elegans* PacBio sequencing data using software Fastq-sample to 20×, 30×, 40×, 50×, 60×, 70×, 80×, 90×, 100× and 150× coverage, together with Hi-C data, and performed de novo assembly independently. Preliminary assemblies were generated using Canu, Flye, Mecat2, Miniasm and Wtdbg2 with raw and corrected reads. A detailed comparison between the resulting assemblies can be found in Fig. 5 and Supplementary Table 4. This study revealed two interesting findings. Firstly, the gap-free de novo assembly is not a suitable option when the data coverage is less than 40X due to the limitation of current de novo assembly tools. As a consequence, GALA switches to gapped assembly for this scenario. Secondly, without Hi-C for scaffolding, Flye and GALA reach the performance curve plateau at 60× and 40× coverage, respectively, regarding the number of scaffolds and N50 of their assemblies. When Hi-C data are applied, the performance curve plateau starts from 40X for Flye and GALA (Fig. 5a, b). The higher

coverage leads to better assembly for Flye with or without Hi-C data by lowering down the number of big gaps and mis-assemblies; however, no notable effects on N50 and the number of scaffolds are observed (Fig. 5c). Thus, the higher coverage of data has no notable effect on GALA assembly in general.

The performance of GALA, as well as almost all assembly software tools, changes significantly with raw-read length and sequencing error. Note that the above analyses are based on the PacBio sequencing data generated with PacBio RSII. Consequently, the lengths of the raw reads are notably smaller and sequencing error is significantly higher than the current PacBio Sequel II. In practice, the sequencing length distribution often varies significantly between different sequencing platforms, genome centres, and sample preparation. Therefore, it is difficult to set a straightforward threshold value for the minimum coverage of data for GALA assembly. As a rule of thumb, GALA can produce gap-free assembly from 25× coverage of PacBio Sequel II data or Nanopore MinION data if N50 of the raw data is larger than 20 Kbp. For PacBio HiFi, 20X coverage works well for GALA due to its low sequencing error rate.

We next evaluated the impact of the number of preliminary assemblies and their quality on the performance of raw reads separation using *O. sativa* Nanopore dataset. The minimum number of preliminary assemblies required by GALA is three. We tested all combinations of 3, 4, 5, 6 and 7 preliminary assemblies with two strategies: an exhaustive test of all combinations and a selective test of all combinations that contain the best preliminary assemblies in terms of N50. After removing organelle genomes, the combinations of three preliminary assemblies in the exhaustive strategy generated on average 25 scaffolding groups, while the selective strategy only generated on average 17 scaffolding groups with much lower variance. With an increase of the number of preliminary assemblies, the difference between the exhaustive and the selective strategy goes down and finally all achieved 14 scaffolding groups (Fig. 5d). The final 14 scaffolding groups represent 10 chromosomes and 4 arms of chromosomes 2 and 11. Since all preliminary assemblies failed to assemble the centromeric regions of chromosome 2 and 11, GALA cannot reduce the number further without using extra information. Overall, the performance of reads separation (CCM) relies on the number and the quality

of preliminary assemblies, the genome complexity, and algorithm's diversity of preliminary assemblies, as well as the quality of sequencing data. As a rule of thumb, 5–10 preliminary assemblies from tools with different graphs or algorithms is practically sufficient for GALA.

### Effect of chromosome-by-chromosome assembly on the assembly graph

We investigated why GALA achieved complete assembly while existing assembly software tools had failed. We postulated that the chromosome-by-chromosome assembly strategy had played a role, and thus, we compared our assembly of *C. elegans* to the regular assembly from Miniasm. This comparison revealed a much simpler computer graph in the chromosome-by-chromosome assembly. In terms of the number of overlaps between reads (graph edges) in the assembly of *C. elegans*, the whole genome assembly generated 190,936,281 edges, whereas the chromosome-by-chromosome assembly only generated 138,678,842 edges (27.37% less). A comparison between the whole genome and the chromosome-by-chromosome assembly is depicted in Fig. 6.

The advantage of chromosome-by-chromosome assembly is more obvious in the regions which contain highly similar sequences, but still have unique markers, e.g., regions with ancient transposons (Fig. 6). In addition, the regions which contain repetitive sequences, but are expanded by long reads, usually allow for a complete assembly by overlap graph-based algorithms, such as Canu or Mecat. However, such assembly is too challenging for *de Bruijn* graph-based algorithms like Wtdbg2. In both scenarios, the GALA method can obtain superior results (Supplementary Fig. 12).

### Discussion

Here, we have presented GALA, a scalable chromosome-by-chromosome assembly method implemented through a multi-layer computer graph. Compared to existing state-of-art assembly workflows and computational tools, GALA improved the contiguity and completeness of genome assembly significantly. Furthermore, our method is highly modular. In detail, the mis-assembly detection module (MDM) should be applicable for error correction regardless of the specific algorithm used for assembly and the contig-clustering module (CCM) can be widely applied for generating consensus assembly from multiple sequences. Although we have focused on de novo assembly in this paper, the modules in GALA should also work equally well in other applications, such as polishing an existing assembly.

In this study, we generated chromosome-scale gap-free assemblies in our experiments. In certain circumstances, we failed to assemble challenging regions such as certain regions in the human genome. This failure is mainly due to the absence of raw sequencing data in these regions as illustrated in Supplementary Fig. 13 and confirmed with external sequencing data, and thus, also occurred in most of the other commonly used computational tools[36–39]. The strength of GALA comes from the multi-layer computer graph model, which is highly flexible in incorporating heterogenous information. As clearly demonstrated in the assembly of the *C. elegans* genomes, combinatory analyses of PacBio and Nanopore sequencing data were achieved.

The performance of GALA also reflects the advantage of chromosome-by-chromosome assembly. Notably, the concept of chromosome-by-chromosome assembly was successfully tested on genome assembly in wheat, for which expensive devices and time-consuming procedures have had to be applied[40,41]. GALA is the first method to demonstrate that this can be achieved computationally. The concept of chromosome-by-chromosome assembly can also be applied to existing computational tools to refine an existing assembly. In addition, scaffolding group-based assembly provides a flexible framework for GALA to support haplotype assembly in the future. This can be achieved by updating the scaffolding group assembly module (SGAM) to support haplotype assembly tools.

Finally, there is still room to improve GALA's assembly quality. Specifically, GALA assembly sometimes collapses long repetitive regions (Supplementary Figs. 5 and 7). In this context, only a single missing read from GALA's Chromosome X scaffolding group when compared to the golden standard, indicating that a bottleneck for the performance of GALA is the scaffolding group assembly module (SGAM) which relies on existing assembly tools. Thus, a new tool that can fully exploit the chromosome structure and depth-of-coverage, similar to centroFlye[42] but applicable to all long repetitive fragments, would be helpful in the future.

## Methods

### Reciprocal alignment between preliminary assemblies

Minimap2[43] (-x asm5) was used to map preliminary assemblies against each other. The raw and corrected reads were aligned to an assembly using BWA-MEM[44] with default parameters.

### Mis-assembly detection module (MDM)

For a genome with $n$ preliminary assemblies, an $n$-layer graph was built by encoding the information from various preliminary assemblies and raw reads. Each layer represents a preliminary assembly $D_x, x \in 1 \ldots n$. Assuming the preliminary assembly $D_x$ contains $m$ contigs, the $j$-th contig in $D_x$ is denoted as $C_j^{D_x}$.

The starting point of the MDM was the reciprocal alignment of $D_x, x \in 1 \ldots n$. Here $n*(n-1)$ alignments were processed since alignments are a bit different if the query sequence and the target sequence are exchanged in the reciprocal mapping, especially for those contigs with multiple hits. We filtered the mapping results based on four criteria: (I) mapping quality, (II) the contig length, (III) the alignment block length and (IV) the percentage of sequence identity. All parameters are tunable in GALA and usually determined according to the reads N50, base accuracy of the preliminary assemblies. The parameter values of the mapping quality and the block length are chosen based on the rate of the false alignment. We recommend increasing the default values for both parameters if ultra-long reads are used or the quality of the preliminary assemblies are better.

We then created inter-layer edges to link nodes between different layers by retrieving the information from reciprocal alignment. Assuming that the $j$-th contig in query layer $D_x$, denoted as $C_j^{D_x}$, is mapped to a set of nodes in layer $D_i, i \in 1 \ldots n, i \neq j$, denoted as $(\{C_j^{D_x}, C_1^{D_1}\}, \ldots, \{C_j^{D_x}, C_i^{D_1}\}, \ldots \{C_j^{D_x}, C_i^{D_n}\})$, a discordance at region $M$ occurs if and only if contig $C_i^{D_k} \in \{C_1^{D_1}, \ldots, C_i^{D_1}, \ldots C_i^{D_n}\}$ is partially mapped to $C_j^{D_x}$ as exemplified in Fig. 2a. Two sequences are partially mapped if they cannot be merged together but their substrings, usually from one end, can be merged together.

Let $N_A$ be the number of contigs partially mapped to $M$, $N_B$ the number of contigs with complete alignment, and $N_S$ be the number of contigs starting or ending at $M$. We considered $M$ as a genuine mis-assembled locus if any condition below is satisfied:

$$N_A \geq (n/2) \tag{1}$$

$$N_B = 0 \ \& \ N_A \geq 2 \tag{2}$$

$$N_A \geq 2 \ \& \ \left(\frac{N_B}{N_A}\right) \leq 0.5 \tag{3}$$

$$N_S > 0 \ \& \ \left(\frac{N_B - N_S}{N_A}\right) \leq 0.6 \tag{4}$$

If a mis-assembly is identified, the node is split into two nodes from the region $M$. This procedure iterates until the whole graph is free of mis-assemblies.

## Contigs clustering module (CCM)

The multi-layer computer graph output by MDM was expanded by adding into an extra layer representing the raw reads. Inter-layer edges are created for nodes in this layer according to their alignment to one or several selected preliminary assemblies. So far, within each layer, nodes were separate from each other and no intra-layer edge existed. If Hi-C information or a genetic map is available, intra-layer edges can be created between two nodes if they are linked together by the Hi-C alignment or are from the same chromosome by the genetic map.

We next browsed through the multi-layer graph node-by-node. For node $C_j^{D_x}$ and its linked inter-layer nodes $\{C_0^{D_1}, \ldots, C_i^{D_1}, \ldots C_i^{D_n}\}$, CCM started by traversing all $\{C_0^{D_1}, \ldots, C_i^{D_1}, \ldots C_i^{D_n}\}$. The nodes that can reach each other through intra- and inter-layer edges are clustered together to form a scaffolding group (Supplementary Fig. 1). In implementation, testing of whether two nodes are reachable to each other through intra- and inter-layer edges could be time-consuming. It can be sped up by artificially creating intra-layer edges between two nodes in the same layer when they are tested to be mutually reachable.

In the previous step of MDM, only contigs with a length larger than a certain threshold value were encoded into our computer graph. Thus, those with smaller sizes were not used for mis-assembly detection. To avoid the situation where unique sequences could be missed out by accident, we here also classified them into existing scaffolding groups for further analysis.

CCM could also be performed in an iterative mode together with the scaffolding group assembly module (SGAM) as demonstrated in the *C. elegans* genome assembly.

## Scaffolding group assembly module (SGAM)

The multi-layer computer graph constructed by CCM can output raw reads in each scaffolding group. The reads within a scaffolding group were subsequently assembled using existing assembly tools, e.g., Flye, Mecat, and Miniasm. In most cases, the assembly tool can produce a gap-free chromosome-scale assembly. We noticed that when a single continuous contig cannot be achieved for a scaffolding group, the breakpoint usually contains a very long repetitive sequence (most of the time in centromeric regions). SGAM provides a simplified version of the overlap graph-based merging algorithm to merge contigs if necessary, as illustrated in Supplementary Fig. 14. However, this procedure sometimes causes collapsing of repetitive regions.

The long repetitive regions could also confuse existing assembly tools in a similar way. When assemblies from multiple software tools are significantly different in terms of length of sequence, we suggest the user to align the raw reads to different assemblies and examine the distribution of the depth-of-coverage. The user should select the best assembly by taking into account the number of assembly errors as well as gaps.

## *Caenorhabditis elegans* assembly

The PacBio dataset contains three different runs and there was a clear batch effect with the sequencing quality and the amount of data between runs. We thus tested the assembly tools with either all runs (290× in coverage), or the biggest run alone (240× in coverage). We also used the reads-correcting-and-trimming module from Canu 1.8[4] to correct the raw reads if the assembly tools take corrected reads as input. Preliminary assemblies were generated using Canu 1.8, Mecat2/Necat[3], Flye 2.4[5], Miniasm 0.3-r179[45] and Wtdbg2[46], from PacBio raw and corrected reads as well as Nanopore raw reads. By comparing the summary statistics of preliminary assemblies, ten preliminary assemblies were chosen for GALA.

GALA modelled the preliminary assemblies and raw reads into 14 independent scaffolding groups. Seven of them were short continuous contigs and the others represented individual chromosomes or chromosome arms. Further analyses by blasting the seven short contigs in the NCBI database indicated that three of them were from *E. coli* contamination and one from the *C. elegans* mitochondrial genome, and thus, were excluded from the subsequent analyses. Of the remaining three short contigs, two of them can be reliably put into the seven previously created scaffolding groups with the help of the assembly of Nanopore reads with Miniasm. Note that to achieve this, we relied on prior information on the number of chromosomes and the distribution of telomeric motifs in *C. elegans*. (Supplementary Fig. 3).

We assembled seven scaffolding groups with SGAM, each into a continuous sequence. Among the seven continuous sequences and one unanchored short contig, four of them revealed the telomere repetitive motif at both terminals, indicating they are complete assemblies of single chromosomes. One chromosome-scale sequence had a telomere repetitive motif at one end, and its missing telomeric repetitive motif can be identified in the unanchored short contig, indicating they both should be merged as a single scaffolding group. The remaining two had a telomere repetitive motif at either side and their sizes clearly indicated they were two arms from a single chromosome. We thus pooled their scaffolding groups together. Finally, we re-assembled the two newly created scaffolding groups and were able to create complete sequences for the two chromosomes with a telomeric repetitive motif at both terminals. Further analyses indicated that the split of this single chromosome into two scaffolding groups in the first run was mainly due to several tandem repeats.

## *Caenorhabditis elegans* genome assembly polishing and quality control

For a more accurate comparison, we polished our assembly with PacBio and Illumina sequencing data. For this purpose, we first ran racon[47] with corrected PacBio reads. The assembly was then polished using quiver 2.3.2[48] with Pbmm2 1.1.0 as an aligner. Finally, we ran pilon 1.23[49] using Illumina sequencing data to correct short errors, especially those in homopolymeric regions.

We evaluated the completeness of our polished assembly with Busco 3.0.0, and compared it to the published assembly, which is also polished using the same Illumina sequencing data as well as the reference genome (Table 1 and Supplementary Fig. 15). We also aligned the Illumina short reads to our assembly using BWA-MEM and called the variants using BCFtools 1.9. Finally, we collected the statistics and compared them to those from the published assembly as a benchmark for the precision of the assembly.

The assembly statistics were calculated using the script at https://github.com/mawad89/assembly_stats. For depth profile evaluation, we mapped the PacBio dataset to 3 preliminary assemblies from Flye, Necat and Miniasm in addition to N2 reference, VC2010 draft assembly and the GALA assembly using Minimap2. Then, we collected the depth information using samtools and mark all the regions with a depth larger than double average depth as collapsed regions. Finally, we used python to plot the average value of window size 5000 bp.

## The effect of the sequencing depth of *Caenorhabditis elegans* on the performance of GALA

We used Fastq-sample to generate 10 sample datasets from the PacBio reads of coverage 20×, 30×, 40×, 50×, 60×, 70×, 80×, 90×, 100× and 150×. Then we used Canu, Flye, Mecat2, Miniasm, and Wtdbg2 to carry out the preliminary assemblies. For each dataset, a Hi-C draft was generated by SALSA2 using the best preliminary assembly and the alignment of the Hi-C sequencing data[19]. For comparison, we assembled each dataset using GALA with or without the Hi-C alignment. The number of mis-assemblies and gap length are collected by comparison mapping to N2 reference genome using (Minimap2 -x asm5). The assembly statistics were calculated using the script at https://github.com/mawad89/assembly_stats. The performances difference is shown in Fig. 3.

### Oryza sativa assembly

Nanopore raw reads was corrected using the Canu 1.8 correcting and trimming module. We found that Necat showed good assembly performance using its own read correction module with the minimal length of reads parameter set at 5000. Eight preliminary assemblies were generated using Canu, Flye, Necat, Miniasm and Wtdbg2 from corrected and raw reads. In SGAM analysis, reads were mapped to Canu and Necat preliminary assemblies. The assembly statistics and the depth profile analysis were conducted in the same way as described in the previous section.

### Human genome assembly

We used the published de novo assembly by HiCanu[28] (3.28GB overall), the current human reference genome GRCh38.p13 and a high-fidelity (HiFi) long reads dataset[28] to assemble the Human genome. A detailed assembly pipeline is available on GALA's page on the github at https://github.com/ganlab/GALA/blob/master/human_genome_pipeline/Pipline.sh.

To determine the length of gaps in Chromosome X, we mapped the scaffolds corresponding to Chromosome X in GALA assembly to the Chromosome X of T2T[30] using minimap2 (-x asm5) and extract the gap length from the paf file.

To compare the GALA performance to the reference-guided scaffolding approach, we performed a reference-guided scaffolding of the published HiCanu assembly[28] using the default parameters of Ragoo[34] and the reference genome GRCh38.p13. Then we used PBJelly[35] to close the gaps with the following parameters for blasr: -minMatch 8 -sdpTupleSize 8 -minPctIdentity 75 -bestn 1 -nCandidates 10 -maxScore −500 -noSplitSubreads. The assembly statistics and depth profile analysis were conducted in the same way as described in the *C. elegans* genome assembly.

### Reporting summary

Further information on research design is available in the Nature Portfolio Reporting Summary linked to this article.

## Data availability

Genome assemblies generated by GALA in this study have been deposited at https://doi.org/10.5281/zenodo.6008862. The PacBio, Hi-C and Illumina reads of *C. elegans* used in this study are available at PRJNA430756 and the Nanopore sequencing data are available at PRJEB22098. *Oryza sativa* Dom Sufid Nanopore sequencing data are available at the European Nucleotide Archive PRJEB32431. We downloaded the human dataset from https://obj.umiacs.umd.edu/marbl_publications/hicanu/index.html, https://obj.umiacs.umd.edu/marbl_publications/hicanu/chm13_20k_hicanu_hifi.fasta.gz. The *C. elegans* N2 reference genome is available at GCF_000002985.6. The VC2010 assembly is available at the European Nucleotide Archive (ENA; https://www.ebi.ac.uk/ena) under accession number PRJEB28388.

## Code availability

The source code of GALA is available from github at https://github.com/ganlab/GALA under the MIT license. The version of the source code of GALA used in the study is available at https://doi.org/10.5281/zenodo.4674388. External software used in the current study were downloaded from the following URLs: assembly_stat.py commit daf6f29, https://github.com/mawad89/assembly_stats; Bcftools1.9, https://github.com/samtools/bcftools/releases; Busco 3.0.0, https://busco-archive.ezlab.org/v3; BWA 0.7.15-r1140, https://github.com/lh3/bwa; Canu 1.8,Canu 2.1 (Hicanu), https://github.com/marbl/canu; Fastq-sample b9a7f71 https://github.com/fplaza/fastq-sample; Flye 2.4, https://github.com/fenderglass/Flye; Hifiasm 0.5-dirty-r247, https://github.com/chhylp123/hifiasm; MECAT2 1.3, https://github.com/xiaochuanle/MECAT2; Miniasm 0.3-r179, https://github.com/lh3/miniasm; Minimap 0.2-r124-dirty

https://github.com/lh3/minimap; Minimap2 2.17-r941, https://github.com/lh3/minimap2; NECAT 0.01, https://github.com/xiaochuanle/NECAT; PBJelly commit d4d2b10, https://github.com/esrice/PBJelly; pbmm2 1.1.0, https://github.com/PacificBiosciences/pbmm2; pilon 1.23, https://github.com/broadinstitute/pilon/releases; quiver 2.3.2, https://anaconda.org/bioconda/genomicconsensus/2.3.2/download/linux-64/genomicconsensus-2.3.2-py27_3.tar.bz2; Racon 1.3.1, https://github.com/lbcb-sci/racon; Ragoo 1.1, https://github.com/malonge/RaGOO;SALSA2 commit 974589f, https://github.com/marbl/SALSA; and Wtdbg2 2.5, https://github.com/ruanjue/wtdbg2.

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

## Acknowledgements

We thank M. Tsiantis for supporting the project and together with R. Mott, D. Megahed and S. Laurent for their helpful comments on the work and Yuxia He for technical support. We also wish to acknowledge S. Morishita for sharing the original *C. elegans* PacBio data with us. This work was supported in part by a Max Planck Society core grant to the Department of Comparative Development and Genetics, a grant from the National Natural Science Foundation of China (Grant No. 3217040347), and grants from the National Science Foundation of Jiangsu Province in China (Grant Nos. JSSCRC2021508 and BK20212010). X.G. is supported by Jiangsu Collaborative Innovation Center for Modern Crop Production. M.A. is supported by the International Max Planck Research Schools programme.

## Author contributions

X.G. conceived the project and interpreted the data. M.A. developed the GALA program and analysed the data. X.G. and M.A. wrote the manuscript. The authors read and approved the final manuscript.

## Funding

## Competing interests

The authors declare no competing interests.
