## [Peer Review File · Nature Communications]

GALA: a computational framework for de novo chromosome-by-chromosome assembly with long readsREVIEWER COMMENTS

Reviewer #1 (Remarks to the Author):

Summary: In this manuscript, Awad et al. present a new software tool called “GALA” which uses multiple sources of information in order to generate improved assembly scaffolds. The tool could be useful to the field, but it is very difficult to assess the true benefit of this software with the paucity of details provided in the methods section. Extensive elaboration on the methods behind each algorithm, and a description of methods used to resolve gaps between contigs in a scaffold must be provided. My comments on the manuscript follow in the order in which I encountered them in the text:

In general: It is clear that the authors are scaffolding contigs together and filling in contig boundaries with sequence, but more work must be done to demonstrate the accuracy of the scaffold filling of the algorithm. Demonstration of full length read alignments across filled gap breakpoints and a paucity of detected structural variants would alleviate some concerns that gap filling is not incorporating new errors. It is one thing to stitch contigs together accurately, and this must be demonstrated using several different methodologies. Case in point: the collapsing of repeat regions is a misassembly that is only briefly mentioned in the manuscript in the Discussions. How many missassemblies or incorrectly filled gaps does GALA produce?

Discussions: GALA appears to benefit greatly from consensus comparison of genome assemblies made from different tools. How many assemblies are needed to generate scaffolds of sufficient quality? What recommendations do the authors have for this important stage of the analysis? Have they done analysis to quantify the minimum number of input assemblies to create higher quality scaffolds?

Line 30: I think that the authors need to define the term, “chromosome-scale sequences” here, as this is not typical terminology used in the field. Do they mean “contigs” that span entire chromosomes? Scaffolds? Further definition is required.

Line 44: I do not think that it is entirely accurate to call GALA an “assembly method” as it relies on the consensus output of other genome assembly algorithms. It seems that the major contribution of the workflow is more akin to “scaffolding” as it incorporates orthogonal data in addition to this inter-assembly comparison methodology. This is not a negative on the whole, but it deserves careful scrutiny in the categorization of the tool as an “assembler” vs a “scaffolding tool.”

Line 99: Did the authors manually filter these contaminants or does the algorithm handle this automatically?

Line 102 and line 105: I think that the authors need to be very careful here with their use of the term, "T2T," as I suspect that they have not validated the lengths of centromeres on their chromosome scaffolds. Data must be provided to demonstrate the T2T status of any scaffolds with respect to centromere completion.

Line 117: I'm surprised that the gap lengths were known here. How did the authors determine the size of gaps within the Flye + Hi-C assembly? Through GALA comparative alignment?

Line 215: The terms used to describe the assembly comparison make it difficult to follow. Which assembly is the "chromosome-by-chromosome" assembly? Also, isn't the GALA graph more simplistic by default as it does not have to create read-to-read associations for use in assembling contigs? I don't understand the point of this comparison, nor do I find the presented results very useful to the main point of the manuscript.

Line 266: Several terms in the description of the algorithm are not properly (or clearly) defined in the text. For example, iteration subscripts ("n", "x") would benefit from descriptions.

Line 268: Given the above description of the multi-layer graph on line 63, is there another layer that consists of raw reads with edges that support raw read alignments to each node? Is this information included in the MDM algorithm? If not, why is it not considered as read alignments have traditionally been a good metric for determining boundaries for misassemblies?

Line 269: Why were these defaults chosen? The sequence identity percentage threshold seems arbitrary without further explanation from the authors.

Line 279: The term "L" is not present in the equations in my version of the manuscript. Also, Must all conditions be satisfied to determine if a locus is misassembled?

Line 288: This section is insufficiently detailed. Specifically, how is linkage data (or an absence of linkage data) used to cluster contigs and generate inter-layer edges? I also do not understand what the authors mean when they say, "CCM could also be performed in an interactive mode together with the LGAM..." Is this not a default step in the algorithm? What are the recommended parameters for this modified approach?

Line 303: Again, what approach is used to scaffold contigs within linkage groups? This section reads more like an extension of the results and provides insufficient detail of the algorithm.

Line 338: More methods details are missing from the text. How were assembly comparisons conducted? How were Human genome assemblies compared? How are gaps filled between linkage group contigs?

Line 342: Replace “homomorphic” with “homopolymeric”.

Table 1: Descriptions of the reference genomes used in this comparison should be added in the legend.

Reviewer #2 (Remarks to the Author):

The authors present an interesting approach for improving assemblies, that could potentially be useful across studies, but If I am honest I think the manuscript could flow better, the results and approach more clearly presented and am still unclear to what extent GALA alone improves assemblies. For example, I would have assumed the authors would pick some consistent baseline (an original set of assemblies) and more thoroughly compare their initial (not manually curated) results from GALA to these. However, the authors do quite a bit of manual downstream adjustments of their assemblies then compare these to completely different existing assemblies. They generally don't present standard metrics for comparing assemblies but pick out particular points. A busco result for one, a missing telomere for another etc. This makes it quite difficult to get an idea of the actual benefit of using GALA given its substantial compute overheads. You are effectively generating up to a dozen assemblies to produce just one final assembly, which will not be an insignificant increase in time and compute. As GALA doesn't itself run these initial assemblies that is a considerable cost in time in generating the inputs that needs to be better justified to the authors as worth it. How long does it take for GALA to run? Can other approaches produce similar results with less overheads? So I think the authors need to do a better job of convincing readers that GALA is making a sizeable, robust difference in assembly quality beyond some anecdotes. For example by showing clear, diverse metrics for each of the GALA runs versus the original input assemblies. Quantifying only the differences made by GALA, not including any downstream adjustments. Table 1 is along the right lines, except comparisons should be made to the original assemblies, the metrics provided should be expanded, and these should be provided for each of the assemblies the authors did, not just this one.

Line 73: “usually each representing a chromosome” I don't think can say usually here as will depend on a whole host of factors. Just because it was the case for the three examples presented by the authors I

don't think you can generalise to all users irrespective of their data. 290X Pacbio coverage is not the norm.

Line 89: "As no current assembly tools support pooled sequencing data from Pacbio and Nanopore platforms" see <https://www.nature.com/articles/s41587-020-00747-w>

Line 91: I got confused by what appears to be the different sets of input assemblers used for each species. It appears sometimes the reads were corrected and other times not. Lines such as "to correct the raw reads if the assembly tools take corrected reads as input" at line 318 are not particularly helpful in this regard. Be specific. List precisely which assemblers were used for each of the species and which were run on corrected and/or uncorrected reads. A (potentially supplementary) table may help here.

The authors do quite a bit of post-processing of their GALA assemblies. For example, lines 97-105 cover blasting to remove contaminants, pooling using supplementary nanopore data, merging based on analyses of telomeres and polishing using illumina and pacbio data. Therefore it is difficult to quantify how much GALA itself actually improved the assembly. So the advantages of specifically using GALA should be better quantified.

The authors merge C.elegans contigs because they only have telomeres on one end. But they can only really get away with this because they know how many chromosomes they are expecting. Notably they don't do the same for the human assembly. Because if they did they would have had the wrong chromosome number. So this approach is very much dependent on prior knowledge, which will often not be available, and is therefore not very robust.

Line 103: "Of note is that the integrative assembly of each linkage group generated gap-free T2T complete sequences for all six chromosomes" but you said one of the groups lacked a telomere on one end?

Line 106: "Note that the VC2010 sample is derived from the N2 reference sample" What does this actually mean? Is it the same sample? How is it different?

Line 108: "The evaluation from Busco 3.0.0 indicated that our assembly successfully assembled two more genes". Appears mean than both other genomes but should clarify. Is it actually just two extra or are some in the other genomes not present in this one? i.e. it is the net difference that is 2.

Line 110: "We performed additional analyses to test the performance of our assembly using the Hi-C dataset generated by the same research group" needs a reference

Line 167: "A single contig free of gaps" this doesn't guarantee the contig is correct. Need more proper assessment of assembly qualities as discussed above. GALA assemblies still seem to have regions of unusual coverage for example (e.g. see supp fig 4)

Line 217: Why chose miniasm and not the other assemblers used?

Line 238: This failure is mainly due to the absence of raw sequencing data

Line 253: "we compared the raw reads..." results should really be in the results section.

Line 267: I may be misreading this but is n the number of assemblies? If so then is $n*(n-1)$ not too many reciprocal alignments? In that genome 1 compared to genome 2 is the same as genome 2 to genome 1?

Line 273: "Assuming that a contig in node C in query layer" I thought the contigs were the nodes?

Line 320: "By comparing the summary statistics of preliminary assemblies, ten preliminary assemblies were chosen for GALA" according to what criteria?

Line 365: The github link to GALA doesn't work. Think GALA is meant to be in capitals.

Dear Reviewers,

We are very grateful to you for the constructive suggestions on how to improve GALA and the paper. We have addressed all the issues raised and have modified our manuscript accordingly. Below are the changes we have performed and our responses to the specific comments and recommendations from you. Key points to be addressed were those on the design and the performance of GALA. To address these issues, we have drawn an analogy between GALA and the chromosome flow-sorting technique, performed a detailed comparison among different assemblies before and after postprocessing, and investigated the change of the performance with the number of preliminary assemblies used.

We hope that these revisions improve the paper such that you now deem it worthy of publication in Nature Communications.

Response to Reviewer #1:

Summary: In this manuscript, Awad et al. present a new software tool called “GALA” which uses multiple sources of information in order to generate improved assembly scaffolds. The tool could be useful to the field, but it is very difficult to assess the true benefit of this software with the paucity of details provided in the methods section. Extensive elaboration on the methods behind each algorithm, and a description of methods used to resolve gaps between contigs in a scaffold must be provided. My comments on the manuscript follow in the order in which I encountered them in the text:

In general: It is clear that the authors are scaffolding contigs together and filling in contig boundaries with sequence, but more work must be done to demonstrate the accuracy of the scaffold filling of the algorithm. Demonstration of full length read alignments across filled gap breakpoints and a paucity of detected structural variants would alleviate some concerns that gap filling is not incorporating new errors. It is one thing to stitch contigs together accurately, and this must be demonstrated using several different methodologies.

Response: We are very sorry for the confusion. The gap-free assembly achieved by GALA is mainly due to the fact that chromosome-based data separation can effectively get rid of data interference when building the computational graph of assembly, as demonstrated in Figure 6. The rationale behind GALA is very close to the chromosome flow-sorting (CFS) technique, that is, separating sequencing data by chromosome to avoid data interference in genome assembly. However, the CFS technique is expensive, time-consuming and labor-intensive. GALA serves as a computational implementation with extra computational loads compared to CFA technique. In our experiments with

publicly available datasets, GALA can achieve chromosome-scale read separation. For more difficult cases, GALA might separate sequencing data from a chromosome into several linkage groups and need extra information to guide their merging. To this end, GALA provides a very flexible framework to support the application of Hi-C data or other sources of information to achieve chromosome-scale read separation.

When assembling raw or corrected reads in linkage groups one-by-one, GALA can directly achieve the gap-free telomere-to-telomere contigs using existing assembly tools most of times. For example, 4 out of 6 chromosomes for the *C. elegans* genome and 12 out of 12 chromosomes for the rice genome were *de novo* assembled into single contigs by running existing assembly tools, such as Flye and Miniasm, on raw reads separated by chromosome. Remaining chromosomes are usually assembled by two tools into 2 or 3 overlapped contigs, and the breakpoints are always inside long repetitive regions. Thus, any simple greedy scaffolding strategy can be employed to merge the contigs together without creating gaps. We have added Supplementary Figure 11 to describe the strategy used in GALA for scaffolding and gap filling. This method is very fast, but it sometimes causes collapsing of repetitive regions.

Case in point: the collapsing of repeat regions is a misassembly that is only briefly mentioned in the manuscript in the Discussions. How many missassemblies or incorrectly filled gaps does GALA produce?

Response: In Supplementary table 2, we discuss the total number of collapsed regions in assemblies of *C. elegans* and *O. sativa* by GALA in comparison to the preliminary assemblies. In addition, we show examples of the collapsed regions in the GALA assembly in Supplementary Figure 5 and 7. There are no chimeric scaffolds or incorrectly filled gaps in the GALA assembly for *C. elegans* and *O. sativa* genomes.

Discussions: GALA appears to benefit greatly from consensus comparison of genome assemblies made from different tools. How many assemblies are needed to generate scaffolds of sufficient quality? What recommendations do the authors have for this important stage of the analysis? Have they done analysis to quantify the minimum number of input assemblies to create higher quality scaffolds?

Response: We thank the reviewer for this valuable comment. We performed analysis to quantify the impact of the number of preliminary assemblies on the performance of GALA. Fig 5d shows the performance change with the number of preliminary assemblies used by GALA. We have added more relevant details from Line 245 to 253 in the updated manuscript.

Briefly, there are two main paradigms for *de novo* assembly. The first is *de Bruijn* graph-based assemblers, such as Flye and Wtdbg. The second is overlap layout consensus graph-based assemblers, such as Miniasm, Canu and Mecat. The preliminary assemblies from different computer graph algorithms complement each other the best and therefore are a good starting point for GALA. In principle, three preliminary assemblies are sufficient. However, in practice, our analyses show that the existing assemblers are often very sensitive to the quality of input sequencing data, even different raw reads correcting algorithms have significant impact on the quality of the output of assemblers. For example, for the rice genome, Flye provided two quite different preliminary assemblies when the raw reads are corrected by its own algorithm or by Canu.

Line 30: I think that the authors need to define the term, “chromosome-scale sequences” here, as this is not typical terminology used in the field. Do they mean “contigs” that span entire chromosomes? Scaffolds? Further definition is required.

Response: We are sorry for this confusion. In tradition, the continuous sequence output by an assembler is called contig whereas the gapped sequence output by scaffolding algorithm is called scaffold. We have updated our manuscript with “chromosome-scale contigs” or “chromosome-scale scaffolds” depending on the existence of the gap.

Line 44: I do not think that it is entirely accurate to call GALA an “assembly method” as it relies on the consensus output of other genome assembly algorithms. It seems that the major contribution of the workflow is more akin to “scaffolding” as it incorporates orthogonal data in addition to this inter-assembly comparison methodology. This is not a negative on the whole, but it deserves careful scrutiny in the categorization of the tool as an “assembler” vs a “scaffolding tool.”

Response: We thank the reviewer for pointing this out. GALA is a computational framework for chromosome-by-chromosome assembly since the main benefit of GALA comes from chromosome-based reads separation. According to the suggestion, we use “assembly tool” in the updated manuscript.

Line 99: Did the authors manually filter these contaminants or does the algorithm handle this automatically?

Response: GALA clusters each of these contigs into an independent linkage group. It is up to the researcher to decide whether to keep these linkage groups for further analyses or even to identify them. In our example, we blasted them manually to detect their sources. As the purpose of the study is the genome assembly of *C. elegans*, we filtered them out. If the study focuses on the bacterial genome, we should keep it for further analysis of course.

Line 102 and line 105: I think that the authors need to be very careful here with their use of the term, “T2T,” as I suspect that they have not validated the lengths of centromeres on their chromosome scaffolds. Data must be provided to demonstrate the T2T status of any scaffolds with respect to centromere completion.

Response: We have followed this advice and removed the term from the sentence.

Line 117: I’m surprised that the gap lengths were known here. How did the authors determine the size of gaps within the Flye + Hi-C assembly? Through GALA comparative alignment?

Response: We map the Flye + Hi-C assembly to the reference genome and the GALA assembly. Then, we extract the coordinates of the gaps with the alignments. The unanchored contigs that were mapped to the same regions of the reference genome where gaps were found would confirm further the lengths of the gaps.

Line 215: The terms used to describe the assembly comparison make it difficult to follow. Which assembly is the “chromosome-by-chromosome” assembly? Also, isn’t the GALA graph more simplistic by default as it does not have to create read-to-read associations for use in assembling contigs? I don’t understand the point of this comparison, nor do I find the presented results very useful to the main point of the manuscript.

Response: We are very sorry for this confusion. In the updated manuscript, we introduced the concept of “chromosome-by-chromosome” assembly by analogy with chromosome flow-sorting (CFS) technique. The rationale of GALA is separating sequencing data by chromosome to avoid data interference in assembly. Then we can use the existing tools to perform a chromosome-by-chromosome *de novo* assembly. To the best of our knowledge, GALA is the only computational tool that employs the strategy of chromosome-by-chromosome assembly. We believe that it is important to illustrate the advantage in a real case in terms of the complexity of underlying assembly graph.

Line 266: Several terms in the description of the algorithm are not properly (or clearly) defined in the text. For example, iteration subscripts (“n”, “x”) would benefit from descriptions.

Response: We have revised our manuscript and added in the definition for ‘n’, ‘x’ and ‘m’.

Line 268: Given the above description of the multi-layer graph on line 63, is there another layer that consists of raw reads with edges that support raw read alignments to each node? Is this information included in the MDM algorithm? If not, why is it not considered as read alignments have traditionally been a good metric for determining boundaries for misassemblies?

Response: We developed a specific module to determine boundaries of misassemblies using long-reads by extending the current method in GALA. However, after intensive tests, we decided not to explicitly incorporate it into the MDM module. There are two challenges with this approach. The first is that it is very slow compared to the current MDM module. The second is the existence of the mis-joint regions in collapsed regions in some cases. As a consequence, the MDM module needs to exploit the coverage information to mark the collapsed region as mis-joints and spilt the contigs if needed. In practice, it is the key to the performance of the CCM module and LGAM module. After considering these two reasons, we did not add this module to GALA, but the MDM can be extended to support it, especially when using a small number of preliminary assemblies.

We have revised our manuscript to make this clear.

Line 269: Why were these defaults chosen? The sequence identity percentage threshold seems arbitrary without further explanation from the authors.

Response: Sorry for having not explained this in the original submission. The parameters of GALA are decided based on the quality of the input sequencing data. The percentage of sequence identity has been chosen according to the base accuracy of the preliminary assemblies. Therefore, in the case of polished preliminary assemblies, GALA can use a higher identity percentage. On the other hand, for preliminary assemblies with very poor base accuracy or when preliminary assemblies of different strains, cell lines or varieties are used, GALA needs to choose a lower identity percentage to achieve better performance.

In addition, assemblers usually do not generate small chimeric contigs for sequencing data with long reads. Therefore, we use the contig length of 5 kbp as a default for MDM. We recommend increasing this parameter to 10 kbp or even more if ultra-long reads are used. The default values of the mapping quality and the block length are chosen by taking into account of the false alignment.

We have added in a brief explanation in the methods section of our updated manuscript.

Line 279: The term “L” is not present in the equations in my version of the manuscript.

Response: We are very sorry for this mistake and have revised our manuscript accordingly.

Also, must all conditions be satisfied to determine if a locus is misassembled?

Response: No, the MDM considers the locus a true mis-assembly if any condition is satisfied. We have made this clear in the updated manuscript

Line 288: This section is insufficiently detailed. Specifically, how is linkage data (or an absence of linkage data) used to cluster contigs and generate inter-layer edges?

Response: The inter-layer edges are generated based on pairwise comparison. The nodes that can reach each other through intra- and inter-layer edges are clustered together to form a linkage group. We have added in a new Supplementary Figure 1 to make it clearer and avoid confusion.

In addition, the CCM module can exploit existing linkage information. For example, if Hi-C data inform that two contigs belong to the same chromosome, an intra-layer edge will be built by the CCM module. All nodes within other layers connected through inter-layer edges to either of these two nodes will be clustered together as they now can reach to others indirectly.

I also do not understand what the authors mean when they say, “CCM could also be performed in an interactive mode together with the LGAM...” Is this not a default step in the algorithm? What are the recommended parameters for this modified approach?

Response: In a challenging scenario, CCM might not be able to cluster all contigs from the same chromosome into the same linkage group. The user could exploit extra linkage information to help with the assembly. For example, a user can perform lab experiments, for example analysis of BACs or Hi-C to decide whether two contigs come from the same chromosome, then create an edge in the graph to connect two corresponding nodes in the interactive mode. CCM will be able to utilize the new graph to improve linkage group-based reads separation for a better assembly.

Line 303: Again, what approach is used to scaffold contigs within linkage groups? This section reads more like an extension of the results and provides insufficient detail of the algorithm.

Response: Sorry for the confusion. The simple scaffolding strategy used by GALA is described in Supplementary Figure 11. We have revised our manuscript to make it clearer.

Line 338: More methods details are missing from the text. How were assembly comparisons conducted? How were Human genome assemblies compared? How are gaps filled between linkage group contigs?

Response: We have revised our manuscript and added in more details.

Line 342: Replace “homomorphic” with “homopolymeric”.

Response: We are sorry for this mistake and have revised our manuscript according to the suggestion.

Table 1: Descriptions of the reference genomes used in this comparison should be added in the legend.

Response: We have added in the citation for the reference genome and the existing assembly to the legend of table one. We also provided the URLs for both assemblies in the updated manuscript.

Response to Reviewer #2:

The authors present an interesting approach for improving assemblies, that could potentially be useful across studies, but If I am honest I think the manuscript could flow better, the results and approach more clearly presented and am still unclear to what extent GALA alone improves assemblies. For example, I would have assumed the authors would pick some consistent baseline (an original set of assemblies) and more thoroughly compare their initial (not manually curated) results from GALA to these. However, the authors do quite a bit of manual downstream adjustments of their assemblies then compare these to completely different existing assemblies. They generally don't present standard metrics for comparing assemblies but pick out particular points. A busco result for one, a missing telomere for another etc. This makes it quite difficult to get an idea of the actual benefit of using GALA given its substantial compute overheads. You are effectively generating up to a dozen assemblies to produce just one final assembly, which will not be an insignificant increase in time and compute. As GALA doesn't itself run these initial assemblies that is a considerable cost in time in generating the inputs that needs to be better justified to the authors as worth it. How long does it take for GALA to run? Can other approaches produce similar results with less overheads?

Response: We are grateful for the helpful comments. We have made substantial revision to improve the presentation of our method. We added in a detailed comparison among different preliminary assemblies and the original GALA assembly and collected the main standard statistics (Supplementary Table 1).

We have drawn an analogy between GALA and the chromosome flow-sorting (CFS) technique, which separates sequencing data by chromosome to avoid data interference in genome assembly or sequence alignment. The rationale behind GALA is very close to the CFS. The CFS technique is expensive, time-consuming and labor-intensive. GALA serves as a computational implementation with extra computational load comparing to the CFA technique.

The computational overhead of GALA is high currently. The preparation steps, e.g., preliminary assembly and mapping, is especially time-consuming. To help with this, we added Figure 5d to illustrate how many preliminary assemblies are necessary. In a typical genome assembly study, a researcher always run several assembly tools and choose the best assembly according to their own metrics. If this is the case, GALA would not practically increase the computational burden dramatically. In addition, producing of multiple preliminary assemblies can be parallelized. The run of the CCM module and MDM module takes less than 1 min for each assembly in all our study. The final

chromosome-by-chromosome assembly does not account for a large proportion of computational costs, as assembly of reads separated by chromosome consumes much less computational resources when comparing to the straightforward assembly of the whole genome, such as the preliminary assembly. Furthermore, it can be done in parallel.

We found it is challenging to find an alternative with much lower computational overhead. Of course, the overhead can be lowered down significantly for genomes where the reference genome is available as we demonstrated for the human genome assembly. But this requires extra information.

So I think the authors need to do a better job of convincing readers that GALA is making a sizeable, robust difference in assembly quality beyond some anecdotes. For example by showing clear, diverse metrics for each of the GALA runs versus the original input assemblies. Quantifying only the differences made by GALA, not including any downstream adjustments. Table 1 is along the right lines, except comparisons should be made to the original assemblies, the metrics provided should be expanded, and these should be provided for each of the assemblies the authors did, not just this one.

Response: We added in a detailed comparison among different preliminary assemblies and the original GALA assembly without polishing and collected the main statistics for them (Supplementary Table 1). We also provided the comparison among the different assemblies in term of number of collapsed regions by examining the distribution of the depth-of-coverage of raw reads alignment (Supplementary Table 2).

Line 73: "usually each representing a chromosome" I don't think can say usually here as will depend on a whole host of factors. Just because it was the case for the three examples presented by the authors I don't think you can generalise to all users irrespective of their data. 290X Pacbio coverage is not the norm.

Response: We have followed this advice and modified the sentence.

Line 89: "As no current assembly tools support pooled sequencing data from Pacbio and Nanopore platforms" see <https://www.nature.com/articles/s41587-020-00747-w>.

Response: Sorry for this. We have removed this claim in the updated manuscript.

Line 91: I got confused by what appears to be the different sets of input assemblers used for each species. It appears sometimes the reads were corrected and other times not. Lines such as "to correct the raw reads if the assembly tools take corrected reads as input" at line 318 are not particularly helpful in this regard. Be specific. List precisely which assemblers were used for each of

the species and which were run on corrected and/or uncorrected reads. A (potentially supplementary) table may help here.

Response: We are so sorry for this confusion. We have provided the more details about the tool name and dataset used in the assembly in Supplementary table 1.

The authors do quite a bit of post-processing of their GALA assemblies. For example, lines 97-105 cover blasting to remove contaminants, pooling using supplementary nanopore data, merging based on analyses of telomeres and polishing using illumina and pacbio data. Therefore it is difficult to quantify how much GALA itself actually improved the assembly. So the advantages of specifically using GALA should be better quantified.

Response: We agree with the reviewer that performance comparison should be more thoughtful and as fair as possible. According to the referee's suggestion, we added in a detailed comparison among different preliminary assemblies and the original GALA assembly and collected the main statistics (Supplementary Table 1).

The authors merge *C. elegans* contigs because they only have telomeres on one end. But they can only really get away with this because they know how many chromosomes they are expecting. Notably they don't do the same for the human assembly. Because if they did they would have had the wrong chromosome number. So this approach is very much dependent on prior knowledge, which will often not be available, and is therefore not very robust.

Response: We agree with the reviewer that the telomeric motif analysis needs prior information and cannot be applied to certain studies. Here we illustrated the high flexibility of GALA in incorporating heterogenous information. If the number of chromosomes is known and the structure of chromosomes are well defined, GALA can use the telomeric analysis or the similar type of information available to improve assembly. We believe this is a big advantage as expensive and labor-consuming Hi-C data is not needed. In fact, GALA can easily partition all raw reads into their chromosomes without the need of motif analyses if Hi-C data is used, as for the Flye/Hi-C assembly.

In practical genome assembly studies, researchers are often more driven by performance of the final assembly. There are assembly tools which needs prior information for assembly. For example, CentroFlye needs very detailed information, such as the structure of centromere region, the unique repeat sequence to assemble the centromeric region of Chromosome X and 6 [1]. We have made it clear that the application of this method needs prior information in Line 374 in the updated manuscript.

Line 103: “Of note is that the integrative assembly of each linkage group generated gap-free T2T complete sequences for all six chromosomes” but you said one of the groups lacked a telomere on one end?

Response: The motif analyses merged the group lacking a telomere at one end with a small group containing short contigs with a telomeric repetitive motif into a new linkage group. The subsequent de novo assembly of reads in the new linkage group generated a gap-free contig for the complete chromosome. We have revised the sentence to avoid confusion.

Line 106: “Note that the VC2010 sample is derived from the N2 reference sample” What does this actually mean? Is it the same sample? How is it different?

Response: The reference genome of *C. elegans* was constructed in 1998 using two wild-type (N2) strains derived from a single *C. elegans* hermaphrodite selected by Brenner in the 1960s.

Unfortunately, none of them exist today and the current reported N2 strains by different labs had acquired significant genetic changes during propagation (Yoshimura et al. 2019, Genome Res.; Howe 2019, Europe PMC). In 2010 Flibotte et al. announced a reference strain (VC2010) as a subculture of the N2 strain (VC196) obtained from the Caenorhabditis Genetics Center in 2002. The authors reported 871 differences between the derived sequence of VC2010 and the reference sequence (Flibotte et al. 2010, Genetics; <https://cgc.umn.edu/strain/VC2010>). Today, because of the unavailability of the reference strains used to construct the reference genome, the VC2010 strain derived from the N2 strain (VC196) is widely used as a reference strain.

We have revised the sentence to avoid confusion.

Line 108: “The evaluation from Busco 3.0.0 indicated that our assembly successfully assembled two more genes”. Appears mean than both other genomes but should clarify. Is it actually just two extra or are some in the other genomes not present in this one? i.e. it is the net difference that is 2.

Response: GALA assembly has 2 more extra genes. We have modified our text to make it clearer.

Line 110: “We performed additional analyses to test the performance of our assembly using the Hi-C dataset generated by the same research group” needs a reference

Response: We have added in the reference.

Line 167: “A single contig free of gaps” this doesn’t guarantee the contig is correct. Need more proper assessment of assembly qualities as discussed above. GALA assemblies still seem to have regions of unusual coverage for example (e.g. see supp fig 4).

Response: We have provided the depth profile of Chr11 in Supplementary Figure 6, showing a comparable result to T2T assembly.

Line 217: Why chose miniasm and not the other assemblers used?

Response: Miniasm and Flye are the two assemblers used for chromosome-by-chromosome assembly in *C. elegans*. Unfortunately, Flye does not report this information explicitly and we can only extract the information from the Miniasm pipeline.

Line 238: This failure is mainly due to the absence of raw sequencing data.

Response: In genome resequencing by Illumina, a region might not get any read to cover it due to various reasons, such as high GC content and DNA quality. For HiFi data we used the human genome assembly, this happened too, though very rarely. We aligned HiFi raw reads and external Nanopore reads to the assembly as shown in Supplementary Fig. 9. We see that there are no HiFi raw reads in some regions at all but we can find Nanopore reads there. So there is missing data in HiFi. One explanation is the DNA quality. Other possible explanation stems from the fact that the HiFi data processing tool cannot tell a long fragment of tandem repeats from a short fragment sequenced multiple rounds in the Pacbio CCS mode. We have modified our text to make it clearer in Line 279 and 280 in the updated manuscript.

Line 253: “we compared the raw reads...” results should really be in the results section.

Response: We have moved this part to the results section.

Line 267: I may be misreading this but is n the number of assemblies? If so then is $n*(n-1)$ not too many reciprocal alignments? In that genome 1 compared to genome 2 is the same as genome 2 to genome 1?

Response: The mapping results are different if the query sequence and the target sequence are exchanged in the reciprocal mapping, especially for those contigs with multiple hits. We have modified our text to make it clearer.

Line 273: “Assuming that a contig in node C in query layer” I thought the contigs were the nodes?

Response: Yes. We have modified the sentence in the updated manuscript.

Line 320: “By comparing the summary statistics of preliminary assemblies, ten preliminary assemblies were chosen for GALA” according to what criteria?

Response: We take into our account the summary statistics, paradigms of algorithms and datasets.

We excluded the highly similar preliminary assemblies generated from different datasets using the same assembly tool. For example, if we have several Miniasm preliminary assemblies generated from different datasets and 2 or 3 of them gave the same alignment results in reciprocal mapping, we chose one of them as a representative. GALA's script "reformat" has implemented to facilitate this comparison.

Line 365: The github link to GALA doesn't work. I Think GALA is meant to be in capitals.

Response: We are grateful for this suggestion. We did not notice any problem in our tests before. We have updated the link according to the advice.

Reference:

1. Bzikadze, A.V. and P.A. Pevzner, *Automated assembly of centromeres from ultra-long error-prone reads*. Nat Biotechnol, 2020. **38**(11): p. 1309-1316.

REVIEWER COMMENTS

Reviewer #1 (Remarks to the Author):

Summary: In this revision, the authors have added substantial clarifying text that improves the readability of the manuscript. The explanation of the method is much clearer; however, I still believe that some additional validation is required to ensure that the limitations and use-cases for the tool are clearly presented.

Supplementary figures 5,6,10 and 11: I think that the authors misunderstood my initial request in the first round of review. I would like to see more validation of the “gap closing” and “contig merger” approach they use in the manuscript through the use of read alignment metrics. Specifically, the gaps that exist in GRCh38.p13 but were closed in the GALA assembly should be compared to the T2T assembly for consistency. If the gap closure was inconsistent between the GALA and the T2T assemblies, read depth and structural variant profiles should be obtained for these regions to identify a potential mechanism for the inconsistency. This would validate the approach used by the workflow for resolving gaps.

Line 121: I recommend removing “a substantially” from the sentence that begins on this line.

Line 251: You were able to consolidate the *O. sativa* assemblies to 14 scaffolds, but it is not indicated in the text if this count corresponds to the known karyotype of the species. If it does not, why was 14 scaffolds the limit in this case? The answer to this does not need to be discursive, but some hints as to why this happened would be informative to future users.

Figure 5d: The additional analysis is welcome and informative. However, I believe that the term, “linkage group,” is incorrectly used here as that indicates a biologically relevant genetic state. “Scaffolds” is the correct term both here and in the main text, and a dashed line on the figure that indicates the real number of biologically determined linkage groups could help distinguish this to the reader.

Reviewer #2 (Remarks to the Author):

The authors are still not really comparing like with like. For example, for the *C.elegans* assembly they appear to take the GALA output, do some manual rejigging of the data based on teleomere motifs, manually filtering out bacterial contigs etc, then assemble this rejigged data. So the subsequent comparison of metrics to the primary assemblies is not with an assembly of the raw GALA linkage group output, but to the manually curated output. The authors don't seem to do any manual curation of the primary assemblies so this doesn't seem a fair comparison. I don't have a problem with the authors also including the metrics of the manually tweaked assembly but I think they need to include the metrics for an assembly from GALA without the manual intervention. This is important as the authors keep saying GALA produces gap free assemblies. But I am assuming they only have a gap free assembly here because of the manual intervention (as otherwise the chromosomes would still be split across linkage groups). So it is not GALA alone producing the "gap free" assemblies.

The authors talking about how GALA gives gap free assemblies is an example of where I think their claims/language sometimes go too far. For example Line 92: "Detailed comparisons indicated that chromosome-by-chromosome assembly always provides better performance" don't say always unless have covered every possible scenario, which the authors haven't. Also line 203 "indicating that GALA's reads separation module works perfectly" don't say perfectly. The authors have already demonstrated it doesn't work perfectly because it took manual curation to merge linkage groups in previous sections of the results.

"Then we added the remaining preliminary assemblies with the best N50 one-by-one. GALA reached the curve plateau with five preliminary assemblies (Fig. 5 D)." It is good to see an assessment of the number of assemblies required but again this is probably not an unbiased result. Obviously to know which assemblies have the best N50 you first need to generate all the assemblies, slightly defeating the purpose of this assessment i.e. to determine how many preliminary assemblies are needed to be generated. It would have been better to randomise this, selecting 3, 4, 5 etc random assemblies multiple times. Then can also add error bars to this plot. The authors say once have the assemblies GALA is quick to run so this should be feasible. Number of linkage groups also seems just one, imperfect metric they are assessing.

Minor points

Line 46 "ten times faster if reads only be aligned to a specific chromosome" should be "ten times faster if reads are only aligned to a specific chromosome"

Line 48 interference should be interfere

Delete "of" at end of line 103.

Line 126 “and is widely used as a substitute to the reference strain” needs citations

Line 159 “compared to the reference genome of Nipponbare” needs citation/accession.

Line 204 Should be “the DXZ1 regions”.

Line 253. Should be “rule of thumb”

Line 304 should be “an n-layer graph”

Line 313 “by taking into account of the false alignment” doesn’t make sense

Line 341 should be “sped up”

Think should be capital L in GALA on line 420

Line 438 Nanopore spelt incorrectly.

Dear Reviewers,

We are very grateful to you for the constructive suggestions to improve the paper. We have addressed all the issues raised and have modified our manuscript accordingly. Below are the changes we have performed and our responses to the specific comments and recommendations from the reviewers. Key points to be addressed were those on the illustration of the performance of GALA by comparing to other assemblies and the reference genomes in an unbiased manner and with convincing examples. To address these issues, we performed a detailed comparison according to the suggestions and presented statistics in the updated figures, supplementary tables and figures. In addition, we investigated the change of the assembly performance with the number and the quality of preliminary assemblies.

We hope that these revisions improve the paper such that you now deem it worthy of publication in Nature Communications.

Response to Reviewer #1:

1. Summary: In this revision, the authors have added substantial clarifying text that improves the readability of the manuscript. The explanation of the method is much clearer; however, I still believe that some additional validation is required to ensure that the limitations and use-cases for the tool are clearly presented.

Supplementary figures 5,6,10 and 11: I think that the authors misunderstood my initial request in the first round of review. I would like to see more validation of the “gap closing” and “contig merger” approach they use in the manuscript through the use of read alignment metrics. Specifically, the gaps that exist in GRCh38.p13 but were closed in the GALA assembly should be compared to the T2T assembly for consistency. If the gap closure was inconsistent between the GALA and the T2T assemblies, read depth and structural variant profiles should be obtained for these regions to identify a potential mechanism for the inconsistency. This would validate the approach used by the workflow for resolving gaps.

Response: We are very grateful for your support of the work and insightful comments. We have added in Supplementary Figure 8 and 9 to demonstrate the performance of gap closing and contig merging by GALA. In Supplementary Figure 8a and b, we show two examples where GALA has allowed closing the gaps in the reference genome. In Supplementary Figure 8c, we show the scaffold produced by contig merging which is the only case in the human genome assembly by GALA. In Supplementary Figure 9, we plot the distribution of the length of gaps in the euchromatin regions in the human reference genome GRCh38.p13 which have been successfully assembled in the GALA assembly when evaluated using the T2T assembly as the benchmark.

Apart from gaps in the GALA assembly that have been detailed in the main text, the remaining inconsistent regions between the GALA assembly and the T2T assembly are all copy number differences of repetitive sequences and most of them are located in centromeric regions and telomeric regions. In Supplementary Figure 8d-f, we show three examples in carton. In Supplementary Figure 10, we plot the length of the centromeric regions in the GALA assembly and the T2T assembly. In addition, the inconsistency can be observed in the read depth plot in Supplementary figure 7.

2. Line 121: I recommend removing “a substantially” from the sentence that begins on this line.

Response: We have followed this advice and removed the words from the sentence.

3. Line 251: You were able to consolidate the *O. sativa* assemblies to 14 scaffolds, but it is not indicated in the text if this count corresponds to the known karyotype of the species. If it does not,

why was 14 scaffolds the limit in this case? The answer to this does not need to be discursive, but some hints as to why this happened would be informative to future users.

Response: The *O. sativa* genome has 12 chromosomes. All preliminary assemblies used by GALA failed to assemble the centromeric regions of Chromosomes 2 and 11. GALA clustered the contigs of preliminary assemblies into 14 scaffolding groups. Among them, 10 groups represent the complete chromosomes, and the remaining 4 groups represent arms of Chromosome 2 and 11. We have modified the manuscript to make this clear.

4. Figure 5d: The additional analysis is welcome and informative. However, I believe that the term, “linkage group,” is incorrectly used here as that indicates a biologically relevant genetic state. “Scaffolds” is the correct term both here and in the main text, and a dashed line on the figure that indicates the real number of biologically determined linkage groups could help distinguish this to the reader.

Response: Thanks for pointing this out. We have been considering this issue and we consulted several experts. “Scaffolds” is an option. Usually, “scaffold” means the sequence has been ordered, oriented and concatenated with “N”. However, in our scenario, there are several contigs in each group and some of them are duplicates from different preliminary assemblies. In addition, for those unique contigs in the same group, we are only sure that they are from the same chromosome but we have not resolved their order and orientation. After careful consideration, we decide to use “scaffolding group” at last.

We have added a dashed line to indicated the real number of chromosomes in Fig. 5d. Thank you for the suggestion.

Response to Reviewer #2:

1. The authors are still not really comparing like with like. For example, for the *C. elegans* assembly they appear to take the GALA output, do some manual rejigging of the data based on telomere motifs, manually filtering out bacterial contigs etc, then assemble this rejigged data. So the subsequent comparison of metrics to the primary assemblies is not with an assembly of the raw GALA linkage group output, but to the manually curated output. The authors don't seem to do any manual curation of the primary assemblies so this doesn't seem a fair comparison. I don't have a problem with the authors also including the metrics of the manually tweaked assembly but I think they need to include the metrics for an assembly from GALA without the manual intervention. This is important as the authors keep saying GALA produces gap free assemblies. But I am assuming they only have a gap free assembly here because of the manual intervention (as otherwise the chromosomes would still be split across linkage groups). So it is not GALA alone producing the “gap free” assemblies.

Response: We are very sorry for the confusion. We have added in a detailed comparison among different preliminary assemblies and the GALA assembly without telomeric motif analysis and manual intervention. As in this circumstance, we failed to assemble 4 chromosomes into the complete gap-free contig, we also provided the chromosome-by-chromosome statistics of our assemblies and compared them to the best preliminary assembly for *C. elegans* and *O. sativa* genomes in Supplementary table 2.

2. The authors talking about how GALA gives gap free assemblies is an example of where I think their claims/language sometimes go too far. For example Line 92: “Detailed comparisons indicated that chromosome-by-chromosome assembly always provides better performance” don't say always unless have covered every possible scenario, which the authors haven't. Also line 203 “indicating that GALA's reads separation module works perfectly” don't say perfectly. The authors have already demonstrated it doesn't work perfectly because it took manual curation to merge linkage groups in previous sections of the results.

Response: Thank you for pointing this out. We have followed the suggestions and changed both sentences.

3. "Then we added the remaining preliminary assemblies with the best N50 one-by-one. GALA reached the curve plateau with five preliminary assemblies (Fig. 5 D)." It is good to see an assessment of the number of assemblies required but again this is probably not an unbiased result. Obviously to know which assemblies have the best N50 you first need to generate all the assemblies, slightly defeating the purpose of this assessment i.e. to determine how many preliminary assemblies are needed to be generated. It would have been better to randomise this, selecting 3, 4, 5 etc random assemblies multiple times. Then can also add error bars to this plot. The authors say once have the assemblies GALA is quick to run so this should be feasible. Number of linkage groups also seems just one, imperfect metric they are assessing.

Response: We have tested all combinations of 3, 4, 5, 6 and 7 preliminary assemblies for the *O. sativa* genome and presented the result in the updated Fig.5d. In addition, we evaluated a subset where the combinations always contain the best preliminary assembly in terms of N50 as an indicator for the effect of the quality of the preliminary assembly as shown in Fig.5d.

Minor points Reviewer #2:

4. Line 46 "ten times faster if reads only be aligned to a specific chromosome" should be "ten times faster if reads are only aligned to a specific chromosome"

Response: We have revised our manuscript according to the suggestions.

5. Line 48 interference should be interfere.

Response: Sorry for this error. We have revised our manuscript.

6. Delete "of" at end of line 103.

Response: We have removed it.

7. Line 126 "and is widely used as a substitute to the reference strain" needs citations

Response: We have added in two references.

8. Line 159 "compared to the reference genome of Nipponbare" needs citation/accession.

Response: We have added in two references and the complete scientific name as mentioned on the references.

9. Line 204 Should be "the DXZ1 regions".

Response: We have added the article in the updated manuscript.

10. Line 253. Should be "rule of thumb"

Response: We have modified the sentence.

11. Line 304 should be "an n-layer graph"

Response: We have added the article in the updated manuscript.

12. Line 313 "by taking into account of the false alignment" doesn't make sense

Response: Thanks for this comment. We have followed this advice and modified the sentence.

13. Line 341 should be "sped up"

Response: We have revised our manuscript by correcting the verb.

14. Think should be capital L in GALA on line 420

Response: We have capitalized the GALA's L in the revised manuscript.

15. Line 438 Nanopore spelt incorrectly.

Response: Sorry for this mistake. We have corrected it in the revised manuscript.

REVIEWERS' COMMENTS

Reviewer #1 (Remarks to the Author):

The authors have addressed all of my major and minor concerns with this revision. The current draft of the manuscript is sufficiently detailed and contains relevant information that will be useful to the research community.

Reviewer #2 (Remarks to the Author):

The authors have largely addressed my previous points. However the new paragraph they added has a few typos etc so have pasted some potential corrected text below

We tested all combinations of 3, 4, 5, 6 and 7 preliminary assemblies with two strategies: an exhaustive test of all combinations and a selective test of all combinations that contain the best preliminary assemblies in terms of N50. After removing organelle genomes, the combinations of three preliminary assemblies in the exhaustive strategy generated on average 25 scaffolding groups, while the selective strategy only generated an average 17 scaffolding groups with much lower variance. With an increase of the number of preliminary assemblies, the difference between the exhaustive and the selective strategy goes down and finally all achieved 14 scaffolding groups (Fig. 5d). The final 14 scaffolding groups represent 10 chromosomes and 4 arms of chromosomes 2 and 11. Since all preliminary assemblies failed to assemble the centromeric regions of chromosome 2 and 11, GALA cannot reduce the number further without using extra information.

Dear Reviewers,

We are very grateful to you for the constructive suggestions to improve the paper during the first and second reviewing rounds. Below are the changes we have performed and our responses to the specific comment from the reviewer #2.

Response to Reviewer #1:

The authors have addressed all of my major and minor concerns with this revision. The current draft of the manuscript is sufficiently detailed and contains relevant information that will be useful to the research community.

Response: We are very grateful for your support of the work and insightful comments during the reviewing process.

Response to Reviewer #2:

The authors have largely addressed my previous points. However, the new paragraph they added has a few typos etc so have pasted some potential corrected text below

We tested all combinations of 3, 4, 5, 6 and 7 preliminary assemblies with two strategies: an exhaustive test of all combinations and a selective test of all combinations that contain the best preliminary assemblies in terms of N50. After removing organelle genomes, the combinations of three preliminary assemblies in the exhaustive strategy generated on average 25 scaffolding groups, while the selective strategy only generated an average 17 scaffolding groups with much lower variance. With an increase of the number of preliminary assemblies, the difference between the exhaustive and the selective strategy goes down and finally all achieved 14 scaffolding groups (Fig. 5d). The final 14 scaffolding groups represent 10 chromosomes and 4 arms of chromosomes 2 and 11. Since all preliminary assemblies failed to assemble the centromeric regions of chromosome 2 and 11, GALA cannot reduce the number further without using extra information.

Response: We are very grateful for your support of the work and insightful comments during the review process. We have followed your corrections and revised our manuscript.